# Online Fall Detection Using Wrist Devices

**DOI:** 10.3390/s23031146

**Published:** 2023-01-19

**Authors:** João Marques, Plinio Moreno

**Affiliations:** 1Instituto Superior Técnico, Unviersidade de Lisboa, 1049-001 Lisboa, Portugal; 2Institute for Systems and Robotics, 1049-001 Lisboa, Portugal

**Keywords:** fall detection, wrist-based dataset, wrist-based solution, machine learning methods, battery/memory limitations, learning version

## Abstract

More than 37 million falls that require medical attention occur every year, mainly affecting the elderly. Besides the natural consequences of falls, most aged adults with a history of falling are likely to develop a fear of falling, leading to a decrease in their mobility level and impacting their overall quality of life. Previous wrist-based datasets revealed limitations such as unrealistic recording set-ups, lack of proper documentation and, most importantly, the absence of elderly people’s movements. Therefore, this work proposes a new wrist-based dataset to tackle this problem. With this dataset, exhaustive research is carried out with the low computational FS-1 feature set (maximum, minimum, mean and variance) with various machine learning methods. This work presents an accelerometer-only fall detector streaming data at 50 Hz, using the low computational FS-1 feature set to train a 3NN algorithm with Euclidean distance, with a window size of 9 s. This work had battery and memory limitations in mind. It also developed a learning version that boosts the fall detector’s performance over time, achieving no single false positives or false negatives over four days.

## 1. Introduction

Every year, approximately 37.3 million falls that require medical attention occur [1]. Falls are the second leading cause of unintentional injury deaths worldwide, the elderly being the most affected group since adults older than 60 years of age suffer the most significant number of fatal falls [1,2]. Moreover, studies [3,4] show that 60% of older adults who have a history of falling are likely to develop a fear of falling, which ultimately leads to a decrease in mobility level, therefore making the individual even more susceptible to future falls.

The main objective of this work is to develop an accurate fall detection mechanism using a commercially available smartwatch. A modern smartwatch is completely detached from the idea of using a medical device, and the wrist location is considered one of the best regarding comfort [5]. The device considered has a 3D accelerometer, gyroscope and orientation sensors, which corresponds to the raw data of this work’s fall detection algorithms, addressing fall detection as a binary classification problem. This work aims to achieve a live fall detector mechanism that presents promising results, meaning that fall occurrences are always detected whilst minimising false positives (FPs). Furthermore, it is intended to provide the community with a new dataset that includes daily activities performed by elderly people since there currently is a lack of data gathered with wrist-based devices for fall detection, especially for elderly people.

This document is structured as follows: Section 2 presents the most common classification algorithms for fall detection and provides the state of the art of fall detection, providing a background on the existing datasets and solutions in the literature; Section 3 provides a description of the dataset this work provides, analyses the data in a machine not limited in resources, and builds the foundations of this article’s solution; Section 4 discusses the fall detector implemented in the watch, its technicalities and limitations; finally, Section 5 draws conclusions from this work and discusses potential future work on the topic of wrist-based fall detection.

## 2. Machine Learning for Fall Detection and Related Work

Fall detection is a binary classification problem in the sense that it consists of a task of classifying the elements of a set into two groups on the basis of a classification rule [6]. Hence, the classification rule can be defined as a procedure by which the elements of a defined population set are each predicted to belong to one defined class [7]. Thus, as the name suggests, binary classification has two classes—a given movement is either considered a fall or not.

As shown later in this article, most works in the literature define different types of falls (e.g., forwards fall, backwards fall). In some of the works, the authors perform fall detection and fall classification, that is, predicting the exact type of fall each movement corresponds to. Fall classification is a multiclass classification problem, meaning that it is a problem of classifying instances into one of three or more classes [8] (in this case, each class corresponds to a different type of fall). Fall classification will not be further explored since it is out of this project’s scope.

As mentioned above, fall detection is a binary classification problem. When judging a movement, it can either be considered a fall or not. The mathematical way to represent whether or not a movement represents a fall is as follows:(1)P(y=1|x;θ).

Equation (1) reads “probability of *y* equals to 1 given *x* parameterised by θ”, where the movement *y* = 1 means “fall” and conversely, *y* = 0 means “not a fall”. In this equation, *x* represents the set of features, and θ represents the parameters that describe how much each feature of *x* affects probability *P*.

The different parameters (θ) are due to the specification of the algorithm used. There are several binary classification algorithms. For fall detection, there are three that are relevant to the present work:K-Nearest Neighbours (kNN)—since elements from the same class tend to have similar characteristics, the intuition behind kNN is to look at the data points around an unknown data point to classify it. Given a training set, with classes “fall” and “not a fall”, the distance of an unknown data point to every other data point is calculated in order to find the *k* nearest points. The data point is then attributed to the class of the majority of the selected neighbours. An odd *k* number should be chosen for binary classification problems to avoid ties in this classification.To estimate the nearest data points, several methods can be used to evaluate the distance between the points, namely Euclidean, Manhattan and Hamming distance, and cosine similarity [9]. The most popular for fall detection is the Euclidean distance. Considering a 2D Cartesian coordinate system, the distance *d* between two points (length of the path connecting them), A$\left(x_{1},y_{1}\right)$ and B$\left(x_{2},y_{2}\right)$, is measured by Equation (2):
(2)$$d=\sqrt{{\left(x_{1}-x_{2}\right)}^{2}+{\left(y_{1}-y_{2}\right)}^{2}}.$$Support Vector Machines (SVM)—briefly, SVM [10,11] classifies a given data point *x* using the decision function D(x), following Equation (3):
(3)class(x)=sign(D(x)).The previous equation attributes the class to a given point with the sign of the decision function, which is computed by Equation (4):
(4)$$D\left(x\right)=\sum_{k=1}\alpha_{k}K\left(x_{k},x\right)+b.$$The coefficients $\alpha_{k}$ are the parameters to be adjusted and the $x_{k}$ are the training patterns, and *b* is the bias. The function *K* is a predefined kernel, usually a linear, quadratic or radial basis function.Decision Tree (DT)—“*A decision tree is a classifier expressed as a recursive partition of the instance space. The decision tree consists of nodes that form a rooted tree, meaning it is a directed tree with a node called “root” that has no incoming edges. All other nodes have exactly one incoming edge. A node with outgoing edges is called an internal or test node. All other nodes are called leaves (also known as terminal or decision nodes). In a decision tree, each internal node splits the instance space into two or more sub-spaces according to a certain discrete function of the input attributes values.*” [12] (p. 165).

It is crucial to evaluate the battery impact of the algorithm considered since a very accurate algorithm could ruin the feasibility of developing a live fall detector given the power constraints. Hence, this work’s objective is to find the algorithm that provides the best accuracy/power consumption tradeoff.

### 2.1. Previous Datasets

In wearable fall detection systems (FDSs), signals are captured from motion sensors to detect falls. There are several places where the user can wear the device: the waist, head, wrist, thigh, chest and ankle. The side of the waist is considered the optimal position for fall detection [13]. Generally, devices closer to the centre of gravity tend to be the most reliable choice for spatial orientation, even though they carry discomfort and appearance concerns.

We analysed several datasets of the state of the art in order to understand their strengths and weaknesses. Wrist-based datasets: (i) Up-Fall [14], (ii) UMA-Fall [15] and non-wrist-based datasets: (i) Mobifall dataset [16], (ii) TFall dataset [17], and (iii) Sisfall [18] were considered. All of the analysed datasets presented limitations that are summarised in the following bullet points:The reasoning for choosing each fall or non-fall movement—it is crucial that the selection of movements is explained, even if the movements chosen are frequent in the literature. Furthermore, the authors of [19] performed a comprehensive survey on how elderly people fall, with 704 women over 65 years old. The results indicate that most falls were caused by trips, slips and losses of balance. The survey highlighted that a fall must not be interpreted alone. Each fall movement has a context—what was happening before the fall (e.g., walking, jogging, sitting)—and a cause (e.g., a slip, a trip, fainting or falling asleep). Datasets where a fall movement starts with an individual standing perfectly still are not only inconsistent with real-life falls but may also inadvertently make researchers take advantage of such perfect conditions to train their algorithms. Moreover, fall movements that do not have a motive of happening (such as a slip) may further remove naturalness from an already unnatural, forced movement.Ideally, the number of individuals executing the actions should be as high as possible. However, due to time and availability constraints, the majority of the studies have low numbers of volunteers executing the actions several times. This approach skews the dataset introducing a bias since an individual is likely to repeat each movement in a more similar way than two different individuals performing the same movement.Young people’s movements are not equivalent to the elderly’s [18,20]. This issue, even though evident, is delicate to resolve. It can be difficult to ensure complete safety for an older person since the risk of injury is always inherent to falling. Results obtained with an algorithm trained with young people’s data must be carefully evaluated since its results would almost certainly deviate when applied to the elderly in real-life falls.Exact conditions of each movement must be documented. The literature lacks detailed descriptions of the exact conditions of each movement, making it hard for researchers to use available datasets.

However, a dataset stood out due to its completion—the SisFall dataset [18]. It compiles data from two accelerometers and a gyroscope placed on an inertial measurement unit (IMU) positioned in the waist of the participants. A total of 38 volunteers participated in this dataset. The authors divided volunteers into two groups: the elder group, consisting of eight males and seven females, aged 60 to 75 years old; the young adult group, composed of eleven males and twelve females, aged 19 to 30. This is the only work in the state of the art with data from older adults, making it considerably more relevant for this work. Naturally, the elder group did not perform falls, except for one man, who due to his expertise in judo, was considered healthy to perform all movements.

This dataset includes 34 different activities—15 falls and 19 activities of daily life (ADLs). The most physically demanding ADLs were not performed by the elderly group. The criteria for choosing each ADL were the following: everyday activities, activities that are similar (in acceleration waveform) to falls, and activities with high acceleration that could generate false positives. This criterion was supported by a survey performed with older adults living alone and with administrative personnel from retirement homes and had results consistent with the ones from the survey conducted in [19].

Lastly, the authors of this work also made available videos detailing the exact conditions of each fall or non-fall activity. This allows addressing the main drawbacks in the literature—the absence of exact conditions in which activities were performed in each dataset, making them a lot more challenging to use.

### 2.2. Previous Fall Detection Systems

Amongst the various fall detection systems, [5,11,14,16,17,21], just one of them works is a wrist-based solution, where kNN achieved an accuracy of 99% on their dataset [5]. However, the authors of [13] state that the side of the waist is the optimal position for fall detection. Generally, devices closer to the centre of gravity tend to be the most reliable choice for spatial orientation, even though they carry discomfort and appearance concerns. Besides being more comfortable when compared to head/chest/waist wearable devices, wrist wearable devices are also less associated with the stigma of wearing a medical device [5,22].

Most of the works in the literature fail to consider fall detection continuously (i.e., a system that runs for long periods of time and evaluates the real-world performance, out of the relatively small dataset) and all the technical, battery and memory limitations that continuous fall detection implies. Instead, fall detection was considered merely as a binary classification problem, and the classification is not done directly in the IMU but rather in a machine not limited in resources (such as a computer). Moreover, FDSs were not confronted with movements external to the dataset and did not evaluate their resistance to false positives and false negatives (FNs).

Ref. [5] was the only work that considered continuous fall detection, with one volunteer participating in six one-hour tests. The best overall result was achieved by the SVM method. Even though all six emulated falls were correctly identified, the number of false positives was far too high to be a usable system in real life. It is stated that the average power consumption of mentioned one-hour tests is 50 mA (this information is incomplete and cannot be compared with the standard energy measure Wh).

This work is the most similar to this project. Given the unpleasant prototype appearance of this solution, the fact that acquired data were not made available, and the poor false positive resistance the online fall detector showed, this work seeks to outperform this work by developing a fall detector with a different architecture, running on a modern smartwatch, whilst developing a complete wrist-based dataset.

## 3. Materials and Methods

### 3.1. Dataset

Given the limitations previous wrist-based datasets showcased, this work proposes a new dataset—WEDA-FALL—Wrist Elderly Daily Activity and Fall Dataset (https://github.com/joaojtmarques/WEDA-FALL, accessed on 10 December 2022). This dataset attempts to provide the community with a complete wrist-based dataset, containing elderly people’s data. The mechanism (https://github.com/joaojtmarques/FitbitGatherDataMechanism, accessed on 10 December 2022) used to compile this dataset is also made available, allowing anyone to gather more data for the dataset.

#### Description of the Datasets’ Movements

The SisFall dataset [18] is likely the most complete study out of every other available dataset in literature. However, it is unusable since it is a waist-based solution, and this work’s goal is to develop a wrist-based FDS. Nevertheless, the dataset this work provides had SisFall as a basis.

The choice of each fall movement of SisFall had a survey as support, whose results pointed to the realisation that a fall usually happens given a context and a cause. Having that in mind, an effort was made to keep each fall direction (forward, backward and lateral) while comprehending the most common/natural scenarios. This work’s selection can be found in Table 1.

The ADL choice had, once again, SisFall as a basis. Table 2 summarises the selection of ADLs for this work. ADL selection was based on the frequency of real-life activities and similarity to fall movements that could generate false positives. The activity choice of SisFall, which attempted to generate false positives, had their IMU position in mind (waist), i.e., movements where the waist would not be stationary. Since this project is focused on a wrist-based solution, the last three ADLs (D09–D11) were strategically chosen and supported by [5].

### 3.2. Description of Participants

The importance of including elderly people’s data in datasets has been mentioned throughout this work. Considering that none of the wrist-based datasets mentioned in Section 2.1 included elderly people’s data, it is one of the main objectives of this work to provide the community with a wrist-based dataset with elderly people’s data.

The groups of participants are, therefore, divided into two: young participants (YP) and elder participants (EP). The statistics of each group of participants can be found in Table 3 and in Table 4, respectively.

The EP group is mainly composed of users of *Casas da Cidade—Residências Sénior Lisboa* (https://www.luzsaude.pt/pt/unidades-luz-saude/casas-da-cidade-residencias-senior/lisboa, accessed on 10 December 2022) Retirement Home. Each participant from this group was carefully chosen with the help of *Casas da Cidade*’s physical therapists. Every elder was healthy and was able to perform each activity selected for them.

The activities each elder individual performed are detailed in Table 5. Note that elder participants were not asked to perform any fall or ADL activities that could harm the individual since it is impossible to assure their safety in these types of movements.

#### 3.2.1. Dataset Acquisition Conditions

It is important to denote that every fall movement was performed on a mattress to avoid the risk of injury. This work’s dataset is accompanied by videos (https://drive.google.com/drive/u/3/folders/1jB3W_sd4-aXkVRsRIdlSLDNdfaXizZ4a, accessed on 10 December 2022) detailing the exact conditions of each movement to solve this significant drawback in the literature.

YPs were asked to repeat each activity three times. Fall F08 was asked to be repeated 4 times, where the first two volunteers would fall towards the side of the watch, while in the last two, they would fall to the opposite side. From YPs alone, this dataset totals 350 (14×7×3+14×4) fall signals, and 462 (14×11×3) ADL signals. The number of repetitions each elder was asked to perform varied a lot, based on each volunteer’s mobility, comfort and fatigue. The total EP signals are 157, having a total of 619 ADL Signals.

Both accelerometer, gyroscope and orientation sensor data were gathered in this dataset with a frequency of 50 Hz.

#### 3.2.2. Ethical Approval and Consent to Participate

The Ethics Committee of *Instituto Superior Técnico* approved all the study procedures. This study was also authorized by the *Casas da Cidade—Residências Sénior Lisboa* Retirement Home. The decision to participate in these experiments was voluntary, and each participant had to sign an informed consent as they were participating in this study. The related files to ethical concerns can be read in Appendix B.

### 3.3. Data Analysis

Section 2.2 highlighted that the problem of fall detection still does not have an accurate solution—even though the current state of the art is able to achieve promising results for accuracy, the false positive rate issue and achieving a good trade-off between accuracy/power consumption/performance are still issues far from solved. In the following sections, this project explores different algorithms offline in a machine that is not as limited in resources as a smartwatch using the dataset described in Section 3.1. In these conditions, it will be possible to conclude the best solution to implement on the watch, given its results and potential impact on the battery.

Computation directly on the device proves to have better battery performance than continuously transmitting data to a different device to perform the calculations [23]. Moreover, ref. [24] (a Pebble smartwatch solution) claimed that with a constant stream of data from the smartwatch to the smartphone, the battery only lasted for 17 to 19 h. After the application was modified to detect falls on the watch and only send suspicious data to the smartphone, the battery increased to a range of 50 to 60 h, an increase of around 300%. Even though it is hard to predict how an MLM would behave on a watch, this work will attempt to develop the fall detector directly on the watch.

#### 3.3.1. Feature Selection

Even though several features proved to have great results throughout the analysis of the literature, a set of features stood out due to its very low computational complexity—the FS-1 feature set, composed of maximum, minimum, mean and variance [21].

Thus, the first step of this study was to understand what would be the accuracy of three different MLMs trained on the FS-1 feature set. To do so, accuracy, specificity and sensitivity values were calculated for 5-fold validation for kNN (with euclidean distance, varying the number of neighbours from 1 to 10), SVM (with radial basis and polynomial functions as the kernel) and DT (with Gini, entropy and log loss as splitting measures). The results, detailed in Table 6, were obtained for the YPs alone, having its features obtained from the raw files, of all three sensors.

Although these are still far from the best results in the literature, these results looked promising considering no preprocessing was done and the features were being calculated on the complete, raw files. In a real-life scenario, a fall detector would not consider temporal windows bigger than 30 s, as some files are even bigger than that.

Therefore, it was essential to understand the ideal window size for this project’s fall detector. Moreover, understanding how lowering the frequency (from the initial 50 Hz) would impact the algorithm is also one of this work’s objectives since the lower the frequency, the lower the battery consumption. In addition, it is crucial to understand if it is possible to maintain high accuracy results using less than three sensors’ data since the least number of sensors used, the least the impact on the battery.

#### 3.3.2. Transforming Raw Data to Different Frequencies and Window Sizes

The dataset was acquired with a frequency of 50 Hz. Filters were applied to reduce the data to frequencies of 40 Hz, 25 Hz, 10 Hz and 5 Hz. Note that no new data were acquired, as the data corresponds to the same files with reduced frequencies.

Before understanding how data were transformed into smaller window sizes, it is determinant to define what is an “Actual Fall”. A fall signal, which in this dataset corresponds to movements F01 to F08, varies in length. Some are as short as 8 s long, whilst others are more than 16 s long. However, it is hard to believe one’s fall would take more than 16 s to complete.

As mentioned in [11], falls have four phases: prefall, impact, adjustment and postfall. The same was observed in this work’s dataset falls, as Figure 1 shows. The fall depicted in Figure 1 corresponds to the accelerometer data from signal F03/U14_R01. Even though the file contains 16.5 s of data, the “Actual Fall” only occurs between the 7 and 10 s mark. Therefore, manual labelling of every single fall of this dataset was done to understand the limits of each “Actual Fall”.

Each “Actual Fall” takes on average 3.1037 ± 0.2731 s. The four phases of falls are usually comprised in this 3-s range, whose middle is the greater variation of acceleration, which in Figure 1 is marked as highlighted as the green dotted line.

Therefore, five different window sizes were considered, ranging between 5 and 9 s. Any window lower than 5 s would be too small considering that falls usually take around 3 s. For the 9-second limit, it seemed reasonable not to surpass three times the average fall time (approximately 3 s). Considering, for instance, a 20-second file and a window size of 5 s, data extraction and labelling are done as follows:The first window would occupy the data between 0 s (the first datapoint) and 5 s (the closest data point to the 5-second mark)FS-1 features are then obtained for this window. The window is classified as ADL or as Fall based on the following criteria:−If the initial file was ADL, the window is always considered as ADL.−If the initial file was fall, and if the new window overlaps at least 50% of the “Actual Fall”, movement is considered a Fall. Otherwise, it is considered an ADL.The same would be done for other windows, considering a certain window_jump. For instance, for a window_jump=1, the following window would occupy the data between 1 and 6 s, and so on, until we reach the last possible 5-second window.

#### 3.3.3. Study of Feature, Sensor, Algorithm, Frequency and Window Size Variations

Since the dataset comprises a relatively small number of participants and repetitions of the ADLs and falls, we perform an exhaustive analysis of the best combination of features and parameters of the feature computation.

This work studies the performance of training different MLMs using the low computational complexity FS-1 feature set [21], composed of maximum, minimum, mean and variance. An exhaustive study was done on the best possible combination of feature selection, frequency, window size and algorithm.

Five different frequencies (50 Hz, 40 Hz, 25 Hz, 10 Hz and 5 Hz)—no_of_freq_comb=5—and five different window sizes—no_of_w_sizes_comb=5—are being evaluated. Moreover, 15 algorithms—no_of_alg_comb=15—10 variations of kNN, 2 variations of SVM and 3 variations of DT are considered. Every possible combination of features is also considered for each of these cases. Since there are considered 4 features, for 3 different sensors, there is a total of 12 features. The total number of combinations from varying the feature set is given by Equation (5):(5)$$feature\_comb={}^{12}C_{1}+{}^{12}C_{2}+{}^{12}C_{3}+{}^{12}C_{4}+{}^{12}C_{5}+{}^{12}C_{6}+{}^{12}C_{7}+{}^{12}C_{8}+{}^{12}C_{9}+{}^{12}C_{10}+{}^{12}C_{11}+{}^{12}C_{12}=4095.$$

Therefore, the total number of combinations considered in this project is 1535625, as Equation (6) shows.
(6)total_combinations=no_of_freq_comb×no_of_w_sizes_comb×no_of_alg_comb×feature_comb=1535625.

With this number of combinations, it is possible to search for patterns of accuracy changes given several variations. The following subsections will study these patterns. Note that the study of the following subsections was done with young participants’ data alone. For each combination, accuracy, specificity and sensitivity values were calculated for 5-fold validation. The description and equations relative to these metrics are depicted in Table 7.

#### 3.3.4. Varying Sensors

Figure 2 outlines the accuracy variations by the number of sensors used. It also considers what sensors were used. This figure showcases a smaller portion of the $1.5356\times 10^{6}$ combinations since the usage of 3 sensors accumulates the most significant portion of combinations. From the analysis of the figure, it is possible to conclude that the accelerometer is the most relevant of sensors (notice how the yellow part of each column, correspondent to the more significant chunk of data points, gets closer to Accuracy=1 when the accelerometer is considered). Even when two sensors are used, the column that does not have the accelerometer is reasonably worse. Figure A1 in Appendix A acknowledges the same pattern in both specificity and sensitivity.

For completion purposes, one can observe in Figure A2 in Appendix A the variation of accuracy, specificity and sensitivity using feature variations of all three sensors.

In previous studies on micro-electromechanical system (MEMS)-based gyroscope sensors [25], their power consumption is tens of times greater than the accelerometers. Thus, it further motivates to train this work’s fall detector with only accelerometer data and it validates previous works’ choice of accelerometer-only solutions.

Note that some figures in Section 3.3 have a red line, which corresponds to the accuracy value of the configuration decided to be implemented on the watch as this work’s fall detector. Said configuration will be summarised later in Section 3.3.8.

#### 3.3.5. Varying Algorithms

Figure 3 illustrates the accuracy variation by algorithm. The first clear conclusion is that SVM proves to have the worst performances. Although kNN and DT are very even, kNN outperforms DT for the best combinations. Moreover, kNN’s columns have a greater yellow portion than the ones of DT, indicating its greater consistency. Similar conclusions can be taken regarding specificity and sensitivity, as Figure A3 in Appendix A shows.

Even though 1NN is the best-performing algorithm, using only the nearest neighbour would likely be unreliable and very susceptible to false positives when considering unknown movements. Therefore, 3NN with Euclidean distance is the choice of this work for its fall detector, as it is the following odd number configuration with the highest accuracy values.

#### 3.3.6. Varying Window Size

Figure 4 shows the accuracy by window size variation. Five different window sizes were considered, ranging between 5 and 9 s. Any window lower than 5 s would be too short considering falls usually take 3 s (value obtained from directly labelling falls of the dataset). For the 9-second limit, it seemed reasonable not to surpass three times the average fall time (3 s).

These results are surprising, as the best accuracy scores are undoubtedly the ones with larger window sizes. Specificity and sensitivity follow the same pattern, as Figure A4 in Appendix A shows. The combinations with the largest values of specificity and sensitivity correspond to low accuracy values. Hence, they are not considered to be relevant.

Previous works [11,14,18,21] presented results for windows smaller than 5 s and never considered bigger sizes of windows. It would be interesting to understand how the already great results of these works would be impacted by changing the window size of their algorithms.

Another advantage of having a bigger window is that the fall detection algorithm is called fewer times. For instance, throughout 60 s, a 6-second window algorithm is called ten times, while a 9-second window algorithm is called six times. The less the algorithm is called, the less battery is consumed.

#### 3.3.7. Varying Frequency

Figure 5 depicts what happens to the accuracy of the algorithms as the frequency changes. As expected, the lower the frequency, the lower the accuracy. The same pattern is found with specificity and sensitivity, as Figure A5 depicts in Appendix A. Similarly to the window sizes, the combinations with the largest values of specificity and sensitivity correspond to low accuracy values. Hence, they are not considered to be relevant.

Notice how the accuracy variation is so similar between 40 Hz and 50 Hz. This conclusion supports the [11] realisation that according to the Nyquist–Shannon sampling theorem, a frequency of 40 Hz suffices to detect falls since the maximum frequency of human body movement is 20 Hz (perfect reconstruction of a movement is guaranteed to be possible using double the maximum frequency of the human body movement). The highest accuracy combinations actually are 40 Hz and not 50 Hz.

#### 3.3.8. Results Discussion and Selection of Approach for Pilot

This section works as a digest of Section 3.3 since the study presented motivated this work’s configuration for the fall detector. The conclusions are summarised in the following bullet points:In an attempt to reduce the number of sensors used by the fall detector, Section 3.3.4 acknowledged the accelerometer as the most relevant sensor for fall detection. Moreover, given that the accelerometer power consumption is tens of times less than the gyroscope, this work decided on an accelerometer-only solution.Section 3.3.5 concluded that although 1NN had the overall best results, kNN with only one neighbour may be unreliable. Therefore, 3NN was chosen, as it is the next odd number configuration with the highest accuracy values.The most surprising conclusion of this study concerns window sizes. Section 3.3.6 showed that bigger window sizes provide greater accuracy than smaller window sizes. The window size of the fall detector this work develops will be 9-ss long.Regarding frequencies, Section 3.3.7 inferred the predictable conclusion that lower frequency values have poorer accuracy performances. The accuracy variation is similar to 40 Hz and 50 Hz frequencies. This work decided on using a 50 Hz frequency for the following two reasons:−Given the previous choices, i.e., an accelerometer-only solution, having 3NN as the algorithm with a window size of 9 s, the best performing configuration was with a 50 Hz frequency.−The smartwatch’s performance was not compromised by using a 50 Hz frequency. Although this choice impacted the battery, it was considered to be worth it given the accuracy gain and its likely greater robustness when considering unknown movements.The impact of different frequencies on a day-to-day basis on the fall detection performance is not evaluated in this work, so it may be part of future works.Lastly, all four features of the low computational complexity FS-1 feature set (maximum, minimum, mean and variance) were used to train the algorithm, as it had the best accuracy than any other subset.

Out of the 1535625 total combinations considered, there were four that shared the best accuracy results. Table 8 summarises these combinations and compares them to the configuration chosen by this work for its fall detector, which is highlighted in red. The table also compares the configurations in terms of specificity and sensitivity.

### 3.4. Impact of Personalisation of Data

Medrano et al. [17] studied the impact of personalising data on the accuracy of fall detection. Its implicit assumption is that a fall detector trained with its user-only data would perform better than a fall detector previously trained with other user data. This work also studied the impact of personalisation, and Table 9 summarises its results.

In order to have a direct comparison, all the values in the table were generated for the same configuration summarised in Section 3.3.8: accelerometer-only data, 3NN with Euclidean distance and window size of 9 s with a frequency of 50 Hz. In total, 10 out of the 14 personalised algorithms had a better performance than the all-users model. These results are coherent with the conclusions in [17], supporting the idea that personalising data is beneficial. However, they do not hold for every user, and there is no concrete justification for why that happens. This is related to the bias/variance of personalized classifiers. Note that the standard deviation of the all-users model is on average 10 times lower than the standard deviation of the individual models for nine users. This shows that with the available data, the all-users data produces more robust results. Finally, to see the actual advantage of the individualised classifier, a longitudinal study using the watch is needed. This topic should be studied in future works.

### 3.5. Impact of Filtered Data

Some works in the literature [18,21] preprocessed the data before using it in the fall detector in an attempt to refine the raw data by removing unwanted noise from the acquired signal. By doing so, they can boost the accuracy of their fall detectors.

This issue was also explored in this work. A rolling window filter was applied to the data, where for every five values, the centre value would become the median of the window. By doing so, the data could become resistant to potential outliers that could skew the data, affecting its features and potentially negatively impacting the algorithm’s performance.

Table 10 summarises the impact of filtering data. It compares the solution of this work with and without filtered data and uses filtered data from all three sensors to collect the FS-1 feature set (for each of them) to train the 3NN algorithm. The conclusion drawn from the analysis of this table is that not filtering the data is more beneficial. This is also advantageous since introducing additional preprocessing overhead to the fall detector is unnecessary, as it would have an impact on the device’s battery life.

### 3.6. Impact of Vertical Acceleration

Quadros et al. [5] studied the impact of using vertical acceleration to train their fall detector, achieving promising results. Therefore, this work explored its impact as well. Acceleration was projected on an inertial referential, where the previous z-axis value of acceleration now corresponds to the acceleration projected in the direction of the vertically upward vector, usually aligned with and opposite to the gravity vector.

Table 11 summarises the impact of using vertical acceleration. It compares the solution of this work with normal acceleration, vertical acceleration and vertical acceleration with the gyroscope. Orientation data were not included since the orientation sensor is already used in the calculus of vertical acceleration. The analysis of this table concludes that not using vertical acceleration is beneficial for this configuration.

### 3.7. Impact of Elder Participants’ Data

Falling is an issue that mainly affects elderly people. Understanding how any fall detector behaves with elderly people is crucial because they are its main target. Unfortunately, data from EPs was acquired very late, and it was impossible to include it in any sub-section of Section 3.3. Working with these data should be a priority for future work.

Table 12 showcases the performance of the fall detector with and without elder participants’ data (for both training and test sets). The rows where EPs’ data are considered to correspond to 100% of ADL data. For the fall data points, it was only used data from 9 YPs to not skew the data, i.e., to keep the data balanced, given that the number of signals of EPs’ data is smaller. From Table 12 analysis, one can conclude that the fall detector has an even better performance with EPs’ data. However, one should analyse these values carefully because EPs did not perform nearly half of the ADL movements selected for this work (Table 2).

## 4. Pilot Study and Results

In this section we analyse the software and hardware limitations that narrow down the number of approaches that will work properly on the wrist device, considering the (i) memory, (ii) power consumption, (iii) recursive feature computation and (iv) online continuous running. An initial pilot with the selected approach had many false positives, which lead to a learning version that asks for user feedback when a fall is detected. The learning version reduced considerably the false positive rate, providing a much better user experience.

### 4.1. Device Limitations

Fitbit’s Software Development Kit (SDK) allows for the creation of apps and clock faces (that can run in the background). When a user opens an app, the code in the clock face stops running, to instead run the code of the app. This means, for example, that if the user opens the settings or the exercise app (while he goes for a run), the watch stops detecting falls.

Secondly, Fitbit code runs on the device that the developer has no control over, such as sleep tracking and reminders to the user to keep active. Through the analysis of the Fitbit Community Forum (https://community.fitbit.com/t5/Community/ct-p/EN, accessed on 10 December 2022), it was possible to understand that each app/clock face only has 128 Kb of RAM and 15 Mb of total memory, including its code. Note that a solid reference does not accompany these memory values, just the information available on the forum (https://community.fitbit.com/t5/SDK-Development/Fatal-Jerryscript-Error-ERR-OUT-OF-MEMORY/m-p/3847635/highlight/true#M9986, accessed on 10 December 2022) and verbal confirmation by a developer in Fitbit’s Development official discord (https://cdn.discordapp.com/attachments/929395896150548503/1026412416025772083/unknown.png, accessed on 10 December 2022). He also stated that the device has a lot more memory than these values, which makes sense since there is code running in the device that a developer has no control over. Therefore, external libraries are strongly advised not to be installed on the device. The algorithm chosen for the fall detector must be implemented from scratch without external optimised libraries. Thirdly, multi-threading is impossible in the development of Fitbit’s apps and clock faces. This means that the feature collection and fall detection are done sequentially, as Figure 6 suggests. The developer must keep the time to detect falls (marked in red in the figure) as low as possible because during that period the watch is not able to collect sensor data.

### 4.2. Feature Calculation

The low computational FS-1 feature set used to train a kNN with Euclidean distance for the solution of this work is comprised of the following parameters: maximum, minimum, mean and variance. Each feature was calculated for each axis: x, y and z, which meant that in total, 12 different features were stored for each movement. These features were calculated and updated as a new sensor reading was available, according to the following equations:(7)$$new\_max=\left\{\begin{array}{ll} new\_reading, & \mathrm{if}\;N=0\\ \max(previous\_max,new\_reading), & \mathrm{otherwise} \end{array}\right.$$
(8)$$new\_min=\left\{\begin{array}{cc} new\_reading, & \mathrm{if}\;N=0\\ \min(previous\_min,new\_reading), & \mathrm{otherwise} \end{array}\right.$$
(9)$$new\_mean=\left\{\begin{array}{cc} new\_reading, & \mathrm{if}\;N=0\\ previous\_mean+\frac{new\_reading-previous\_mean}{N}, & \mathrm{otherwise} \end{array}\right.$$
(10)$$new\_variance=\left\{\begin{array}{cc} 0, & \mathrm{if}\;N\leq 1\\ \frac{previous\_variance+(new\_reading-previous\_mean)*(new\_reading-new\_mean)}{N-1}, & \mathrm{otherwise} \end{array}\right.$$

In these equations, new_reading is the newest sensor reading available, and *N* is the counter of sensor readings in the current window. For each window, *N* would range from 0 to 450 (9-second window × 50 Hz of frequency). Equations (9) and (10) are supported by [26,27]. Each one of the listed equations is calculated for each axis.

### 4.3. The Dataset Choice and Need for Learning Version

The analysis of Section 3.3 was done on a computer that does not have limited resources for the amount of recorded data. That allowed us to train and test algorithms with around 5000 data points. However, for the Fitbit Sense, it was possible to conclude that the dataset on the watch could not be bigger than 200 data points due to the RAM and total memory limitations.

There were found two ways to overcome this limitation: firstly, by renaming each metric to a smaller string (e.g., max_accel_x to *a*) so that each file would occupy fewer bytes; secondly, by dividing the dataset into several different smaller files, so that when the code would try to load them into RAM, it would not crash. These optimisations allow the dataset to be decently large, as long as every file combined does not exceed the memory limit. However, additional overhead is introduced in the opening of files that significantly impacts the total time the algorithm takes to complete and the device’s battery life.

The initial fall detector implemented was very simple: once the algorithm recognised a fall, it would vibrate and show a message on the screen saying a fall was detected. At that moment, it was possible to conclude that although the compiled dataset was as broad as possible, it did not include all the daily activities that can be misclassified as falls. Movements such as washing hands that we perform so often during the day would constantly be classified as a fall, giving the user an unpleasant experience. Thus, a new mechanism was introduced—the user-based learning version—to improve the watch’s performance, improve its user experience, and reduce the number of false positives.

This version works as follows: when the algorithm recognises a fall, besides vibrating and showing a message on screen, it also asks for user feedback on whether the fall was, in fact, a fall or not. The watch then adds the user feedback to its dataset so that this movement is considered in the future.

The learning version proved to have immediate results on the performance and user experience of the fall detector, as the false positive rate was reduced significantly. However, the developer must be careful that the dataset does not become imbalanced. One should also consider that the more data points the dataset has, the more iterations it must do, meaning a longer time to detect falls (Figure 6) and overhead, more significant impact on battery and worse overall performance of the device.

### 4.4. Solution Evaluation

In order to evaluate the learning version solution, volunteers were asked to use the watch for the duration of a battery lifetime—100% to 0%. They were encouraged to play with the device as they wanted—they could try falling, try to induce false positives, or just live a normal day. Older volunteers were not encouraged to perform any falls. Table 13 summarises its results.

The initial dataset had around 400 data points. These data points were directly acquired from the dataset described in Section 3.1. It comprised 200 fall and 200 ADL data points. User_1 used the watch for nearly a month before evaluating the solution and achieving the results presented in Table 13. User_1 was very active this month, and by the end, he added nearly 170 data points to the dataset.

When User_1 tested the fall detector, the watch correctly detected three falls (frontal, backward and lateral); it failed to recognise a fall (backward) and wrongly identified four falls during the battery lifetime (33 h). These wrongly identified falls corresponded to rapid movements entering and leaving the bed, taking a shower and shaking clothes before hanging them to dry.

User_5 tested the fall detector immediately after User_1 (after charging the device), meaning that the initial dataset for User_5 already had the feedback from User_1. The same happens for every row in Table 13 as they are ordered chronologically. Similarly to User_1, User_5 simulated three falls (frontal, backward and lateral) which were correctly identified, and the watch also failed to recognise a backward fall. Curiously, User_5 inadvertently fell during the experiment (lateral fall while sitting), and the watch correctly identified it. The watch wrongly identified seven falls, corresponding to movements such as hitting a table, clapping hands, adjusting posture while sitting in a chair, opening a box and removing a bicycle lock.

A few weeks separate the tests of User_5 and User_A. User_1 used the watch in the meantime. During this period, he noticed the watch started to lose performance. He also noticed that the time to detect falls (Figure 6) was close to 2 s, corresponding to more than 20% of the designated window size.

Nevertheless, the watch was given to User_A to test the fall detector. Participants identified with a letter instead of a number were not asked to simulate falls, given their age. However, User_A fell by accident (frontal fall), and the watch correctly identified the fall. The fall detector wrongly detected falls 12 times. It happened when he was driving, clapping his hands, celebrating after his favourite football team scored and sitting back down on his chair right after the celebration.

After the poor results from User_A, it was decided to delete data points from the dataset. At this point, there were nearly 650 data points in the dataset. The first 200 ADL data points were deleted from the very initial dataset, prioritising the broader user-feedback data. An extra 50 datapoints of user-feedback data that corresponded to a period in which User_1 was on holiday and played volleyball—the fall detector wrongly detects many falls while playing volleyball—since these were movements different from the ordinary that could be negatively impacting the fall detector.

User_1 decided then to re-test the fall detector. He simulated three falls (frontal, backward and lateral) which were correctly detected, although one backward fall was not detected once again. The watch wrongly detected two falls when the user went up the stairs very rapidly. in this test, he noticed a significant reduction in the time to detect falls (Figure 6) and an increase in the device’s battery life. The watch was then given to User_B, User_C and User_A (by this order), and the watch did not report a single fall.

Quadros et al. [5], the closest previous work to this work, performed six different one-hour period tests. The prototype presented eight falls in an hour gym session, eight falls in a 15 min bath and 90 s of false alarms during heavy cleaning. They also mention that all six emulated falls in each test were correctly identified. This work, in comparison, only had 25 falls over 238 h (almost ten full days) when the device was tested. However, this solution failed to recognise falls three times, which did not happen in the previous work’s six emulated falls.

This less successful result regarding false negatives is attributed to the nonexistence of multithreading, the limited processing power and the extremely low memory’ limits for Fitbit’s apps and clock faces. This work’s solution will confidently never detect any fall that happens in the time to detect falls (Figure 6). To reduce false positives, this project even considered having multiple windows besides the main 9-second window to “confirm” the fall, which could give robustness to the solution. However, the absence of data in the time to detect falls (Figure 6) period and the already low limits of computational capacity make such approaches impossible.

Regarding the battery, the change in battery life is clear as the watch’s dataset gets bigger or smaller. According to [28], Fitbit Sense’s battery is a 1.02 Wh (266 mAh @ 3.85 V) cell. Considering 34 as the average number of hours of the battery lifetime (from Table 13), this solution’s clockface consumes 0.03 W per hour ($\frac{1.02\;\mathrm{Wh}}{34\;h}$). According to online information [29,30], Fitbit Sense’s battery lifetime usually lasts 6 days and has a charge time (10–80%) of 40 min. It means Fitbit Sense’s factory clockface consumes approximately 0.007 W per hour ($\frac{1.02\;\mathrm{Wh}}{6\times 24\;h}$). Therefore, this solution consumes approximately 0.023 W per hour (0.03−0.007), corresponding to 0.552 Wh.

Regarding user experience, users unanimously claimed it was a good experience. For those who experienced it, the false positives were sometimes annoying, for instance, when User_A was driving. However, this user was utterly surprised by the absence of false positives the second time he tested the fall detector.

On a final note, it is worth mentioning that better performance was acquired with the older volunteers, which is not surprising, as most of their movements are less vigorous than those of younger volunteers.

## 5. Conclusions and Future Work

### 5.1. Conclusions

This work proposes an online wrist-based fall detection system with a modern-day watch, a device completely removed from any unpleasant stigma. We contribute with: (i) a carefully designed dataset and acquisition protocol, and (ii) an online fall detection that gets user labels while running.

This work proposes its own dataset and protocol acquisition that carefully follows the design and execution of the ADLs and falls, which was the main limitation of previous datasets. In addition, it contains ADLs from healthy elderly and young people, as well as falls by young people, making it a unique dataset in terms of ADL. Using this new dataset, we performed comprehensive offline experiments to select the best parameters (i.e., sensors selected, feature computation, learning algorithm and window size), such that the algorithm can run online and the battery consumption of the smartwatch is not compromised. We implemented a fall detector in the watch using the selected offline parameters while adding the capability of getting feedback from the user in case of false positives. The learning version greatly reduces false positives and improves the users’ experience. In seven different battery life-long tests, with five different people, lasting 238 h (almost ten full days), there were only twenty-five false positives and three false negatives detected. In total, 11 out of 14 falls were correctly detected during this period. The average battery life of the device was 34 h, consuming 0.552 Wh.

The number of false positives and false negatives can be attributed to the limited hardware resources. A Fitbit app or clockface only has 128 Kb of RAM and 15 Mb of total memory, including its code, and multi-threading is impossible on these devices. This means that feature collection and fall detection happen sequentially, never synchronously.

### 5.2. Future Work

#### 5.2.1. Study How Other Algorithms and Features Would Perform on the Watch

We did an exhaustive offline study to select the ML algorithm, its parameters and feature computation. However, a more comprehensive study needs to be done on the online implementation to find the parameters of the ML algorithm and features that lead to better classification results. For instance, a fine-grained frequency selection and re-evaluation of the window size of the fall detector.

#### 5.2.2. Understand Impact of Personalisation

We briefly analysed the impact of using personalized data for fall detection. From these experiments, the advantage of a personalised fall detector is not clear. In the future, this issue could be explored directly in the watch, as different users perform movements differently and different sets of movements throughout the day. Understanding whether having a “learning period”, where the fall detector could familiarise itself with the user before evaluating its performance, would increase its results would be an exciting contribution to this topic.

#### 5.2.3. Understand Performance of Fall Detector with Elder People

This work made available the first wrist-based dataset with elder people’s data. However, given this data’s late collection, it was impossible to do an exhaustive analysis. Since falling is an issue that primarily affects the elderly, it is vital to always have elderly people in mind—a fall detector is only as good as its results with elderly people. Therefore, the fall detector should be evaluated with elderly people in future works.

#### 5.2.4. Achieve Improvements by Contacting Fitbit

Smartwatches are compactly designed for weight and reduced size, so that comes with hardware limitations of the device: (i) memory limitations and (ii) no availability of multi-threading.

Since the watch has more memory than the one allocated to each application, Fitbit should be contacted, in order to ask for a better environment. Having more hardware resources would allow optimisations in the software, which would inevitably positively impact the battery, and potentially achieve better accuracy results. 

## Figures and Tables

**Figure 1 sensors-23-01146-f001:**
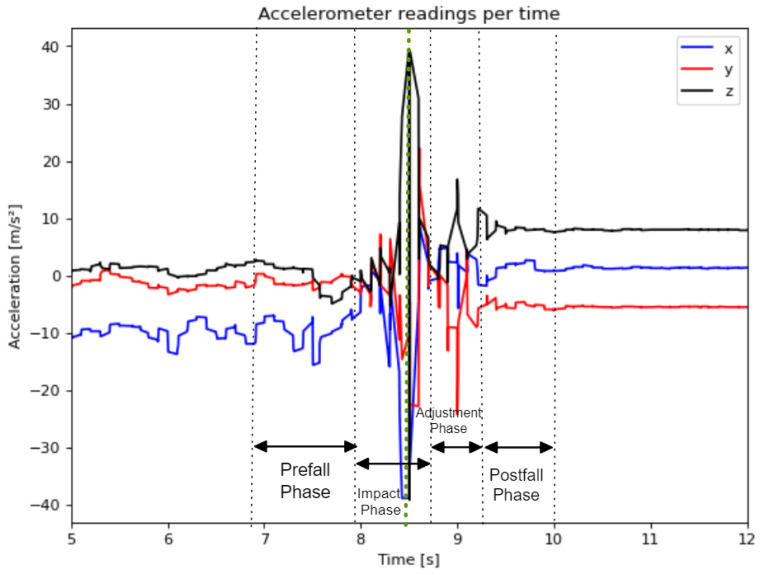
Observed phases of a fall for signal F03/U14_01.

**Figure 2 sensors-23-01146-f002:**
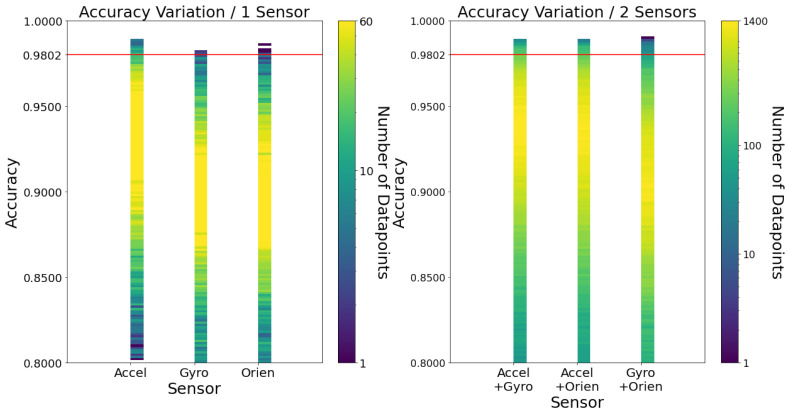
Accuracy variation by sensors. Each data point corresponds to a combination of frequency, window size and learning algorithm.

**Figure 3 sensors-23-01146-f003:**
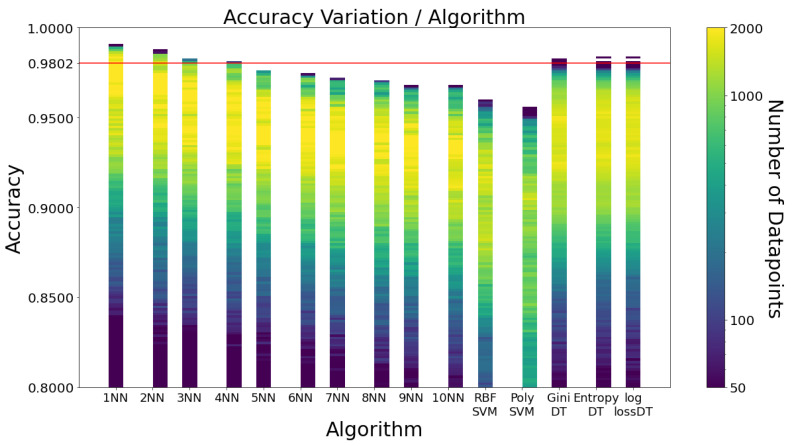
Accuracy variation by algorithm. Each data point corresponds to a combination of frequency, window size and learning algorithm.

**Figure 4 sensors-23-01146-f004:**
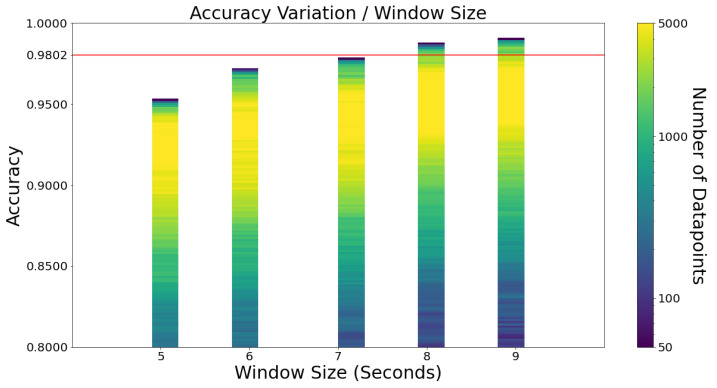
Accuracy variation by window size. Each data point corresponds to a combination of frequency, window size, and learning algorithm.

**Figure 5 sensors-23-01146-f005:**
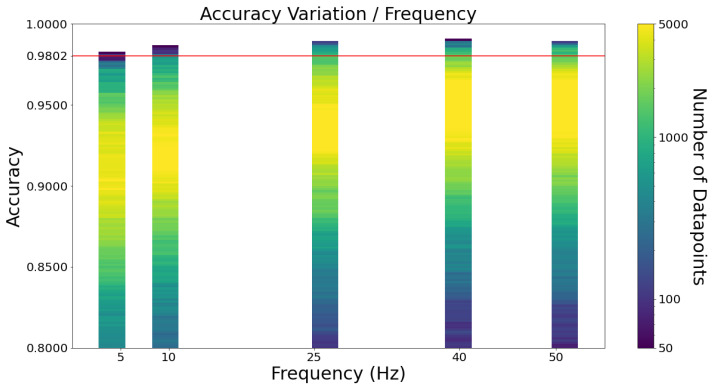
Accuracy variation by frequency. Each data point corresponds to a combination of frequency, window size and learning algorithm.

**Figure 6 sensors-23-01146-f006:**
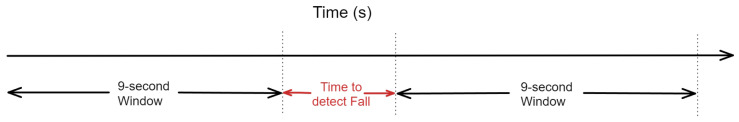
Feature collection and fall detection are sequential.

**Table 1 sensors-23-01146-t001:** Types of falls selected for this work.

Code	Activity
F01	Fall forward while walking caused by a slip
F02	Lateral fall while walking caused by a slip
F03	Fall backward while walking caused by a slip
F04	Fall forward while walking caused by a trip
F05	Fall backward when trying to sit down
F06	Fall forward while sitting, caused by fainting or falling asleep
F07	Fall backward while sitting, caused by fainting or falling asleep
F08	Lateral fall while sitting, caused by fainting or falling asleep

**Table 2 sensors-23-01146-t002:** Types of ADLs selected for this work.

Code	Activity
D01	Walking
D02	Jogging
D03	Walking up and downstairs
D04	Sitting on a chair, waiting a moment and getting up
D05	Sitting a moment, attempting to get up and collapsing into a chair
D06	Crouching (bending at the knees), tie shoes, and get up
D07	Stumble while walking
D08	Gently jump without falling (trying to reach high object)
D09	Hit table with hand
D10	Clapping hands
D11	Opening and closing door

**Table 3 sensors-23-01146-t003:** Statistics of young participants.

User_id	Age	Height (m)	Weight (Kg)	Gender
1	22	1.76	56.3	Male
2	22	1.78	56.0	Male
3	20	1.73	69.5	Male
4	21	1.70	57.1	Female
5	23	1.67	59.6	Male
6	22	1.67	69.0	Male
7	21	1.78	68.1	Male
8	23	1.62	61.0	Female
9	22	1.70	52.0	Female
10	23	1.83	77.0	Male
11	23	1.69	61.8	Female
12	23	1.78	64.5	Female
13	22	1.79	66.0	Male
14	46	1.84	83.0	Male

**Table 4 sensors-23-01146-t004:** Statistics of elder participants.

User_id	Age	Height (m)	Weight (Kg)	Gender
21	95	1.70	71.0	Male
22	85	1.53	62.0	Female
23	82	1.60	60.0	Female
24	81	1.52	63.0	Female
25	81	1.73	72.0	Female
26	83	1.75	85.0	Male
27	89	1.71	71.5	Male
28	88	1.57	52.5	Female
29	77	1.60	65.9	Female
30	80	1.79	72.0	Male
31	88	1.63	53.0	Female

**Table 5 sensors-23-01146-t005:** Activities elder participants performed.

User_id	Activities Performed
21	D01; D04; D09; D10; D11;
22	D01; D04; D09; D10; D11;
23	D01; D03; D04; D09; D10; D11;
24	D01; D03; D04; D09; D10; D11;
25	D01; D04; D09; D10; D11;
26	D01; D03; D04; D09; D10; D11;
27	D01; D03; D04; D09; D10; D11;
28	D01; D03; D04; D09; D10; D11;
29	D01; D03; D04; D09; D10; D11;
30	D01; D03; D04; D09; D10; D11;
31	D01; D03; D04; D09; D10; D11;

**Table 6 sensors-23-01146-t006:** Accuracy of MLMs on YPs’ raw data.

Algorithm	Accuracy Mean	Accuracy stdev	Specificity Mean	Specificity stdev	Sensitivity Mean	Sensitivity stdev
3NN	0.94292	0.02835	0.94128	0.05029	0.94483	0.02248
8NN	0.93988	0.02696	0.94691	0.03385	0.93103	0.05587
7NN	0.93833	0.02193	0.93576	0.03563	0.94138	0.04967
5NN	0.93524	0.02770	0.93576	0.03824	0.93448	0.03738
6NN	0.93524	0.03215	0.94965	0.04424	0.91724	0.05897
1NN	0.93369	0.03127	0.94413	0.03603	0.92069	0.03575
4NN	0.93369	0.03569	0.96084	0.03516	0.90000	0.05770
9NN	0.92756	0.02594	0.91909	0.03076	0.93793	0.05666
2NN	0.92755	0.02483	0.97203	0.02981	0.87241	0.03575
10NN	0.92448	0.02276	0.92187	0.03263	0.92759	0.06022
rbf_SVM	0.92295	0.00956	0.92187	0.03680	0.92414	0.04815
poly_SVM	0.91369	0.01930	0.94695	0.03362	0.87241	0.05796
entropy_DT	0.88754	0.03098	0.89147	0.02636	0.87931	0.04043
gini_DT	0.88594	0.02424	0.89687	0.02154	0.87586	0.02833
log_loss_DT	0.88134	0.04027	0.89421	0.02296	0.86207	0.04223

**Table 7 sensors-23-01146-t007:** Metrics that will be used to evaluate the solution.

Metric	Description	Equation
Accuracy	Number of correctly predicted data points out of all the data points	$\frac{TP+TN}{TP+TN+FP+FN}$
Sensitivity	Proportion of actual positive cases, which got predicted correctly	$\frac{TP}{TP+FN}$
Specificity	Proportion of actual negative cases, which got predicted correctly	$\frac{TN}{TN+FP}$

**Table 8 sensors-23-01146-t008:** Comparison of the highest accuracy configurations and this work’s choice.

Configuration	Features	Accuracy Mean	Accuracy stdev	Specificity Mean	Specificity stdev	Sensitivity Mean	Sensitivity stdev
1NN 40 Hz, 9 s	Accel—Max, Min, Mean Gyro—Min, Variance Orien—Min, Variance	0.98936	0.00662	0.99081	0.00431	0.98757	0.01225
1NN 40 Hz, 9 s	Accel—Max, Min, Mean Gyro—Min, Mean, Variance Orien—Min, Variance	0.98936	0.00662	0.99081	0.00431	0.98757	0.01225
1NN 40 Hz, 9 s	Accel—Max, Min, Mean Gyro—Min, Mean, Variance Orien—Min	0.98936	0.00662	0.99081	0.00431	0.98757	0.01225
1NN 40 Hz, 9 s	Accel—Max, Min, Mean Gyro—Min, Variance Orien—Min	0.98936	0.00662	0.99081	0.00431	0.98757	0.01225
3NN 40 Hz, 9 s	Accel—Max, Min, Mean, Variance	0.98025	0.00463	0.97842	0.00839	0.98249	0.00368

**Table 9 sensors-23-01146-t009:** Accuracy impact given the personalisation of data.

User_id	Accuracy Mean	Accuracy stdev	Specificity Mean	Specificity stdev	Sensitivity Mean	Sensitivity stdev
8	0.99753	0.00552	1.00000	0.00000	0.99500	0.01118
9	0.99630	0.00828	0.99310	0.01542	1.00000	0.00000
2	0.99333	0.00913	1.00000	0.00000	0.98182	0.02490
7	0.99286	0.00978	1.00000	0.00000	0.98261	0.02381
1	0.98987	0.01059	0.98105	0.02807	0.99574	0.00952
11	0.98571	0.02130	1.00000	0.00000	0.93056	0.10111
10	0.98408	0.01671	0.99333	0.01491	0.97000	0.04472
6	0.98290	0.01566	0.97167	0.04652	0.99048	0.01304
4	0.98286	0.02556	0.98519	0.03313	0.97778	0.04969
12	0.98252	0.02622	1.00000	0.00000	0.94182	0.08851
All Users	0.98025	0.00463	0.97842	0.00839	0.98249	0.00368
14	0.97549	0.01489	0.94127	0.03285	0.99565	0.00972
13	0.97401	0.01822	0.96800	0.03347	0.98139	0.02550
3	0.97287	0.02872	0.98261	0.03889	0.94286	0.07825
5	0.97160	0.01693	0.95936	0.04622	0.98421	0.01441

**Table 10 sensors-23-01146-t010:** Usage of filtered data.

Label	Accuracy Mean	Accuracy stdev	Specificity Mean	Specificity stdev	Sensitivity Mean	Sensitivity stdev
No Filtered Data	0.98025	0.00463	0.97842	0.00839	0.98249	0.00368
Three Sensors Filtered Data	0.97820	0.00414	0.97978	0.00787	0.97624	0.00253
Accel-Only Filtered Data	0.97313	0.00414	0.97427	0.00733	0.97172	0.00281

**Table 11 sensors-23-01146-t011:** Usage of vertical acceleration.

Label	Accuracy Mean	Accuracy stdev	Specificity Mean	Specificity stdev	Sensitivity Mean	Sensitivity stdev
Normal Accel	0.98025	0.00463	0.97842	0.00839	0.98249	0.00368
Vertical Accel + Gyro	0.97185	0.00504	0.97197	0.00696	0.97172	0.00776
Only Vertical Accel	0.96577	0.00516	0.96370	0.00893	0.96832	0.00760

**Table 12 sensors-23-01146-t012:** Usage of elder participants’ data.

Label	Accuracy Mean	Accuracy stdev	Specificity Mean	Specificity stdev	Sensitivity Mean	Sensitivity stdev
3 sensors EPs’ data	0.98247	0.00786	0.97108	0.01324	0.98963	0.00655
Accel only EPs’ data	0.98141	0.00860	0.96969	0.01155	0.98876	0.00785
YPs’ data	0.98025	0.00463	0.97842	0.00839	0.98249	0.00368

**Table 13 sensors-23-01146-t013:** Results of the fall detector (rows are ordered chronologically).

User_id	Age	Battery Lifetime	#Falls Correctly Detected	#False Negatives	#False Positives	#Datapoints
1	22	33 h	3	1	4	∼570
5	23	33 h	4	1	7	∼578
A	61	32 h	1	0	12	∼650
1	22	35 h	3	1	2	∼400
B	56	35 h	0	0	0	∼406
C	86	35 h	0	0	0	∼406
A	61	35 h	0	0	0	∼406

## Data Availability

Not applicable.

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
