# Peer review of "Online Fall Detection Using Wrist Devices"

_sensors, 2023, doi:10.3390/s23031146_

Round 1

Reviewer 1 Report

This paper mainly uses the accelerometer of wearing wrist watch to judge whether the elderly fall and collect the data set, and uses the FS-1  feature set to train the 3NN algorithm to improve the performance of the fall monitor.

The fall data collected by the author through the experiment is of great reference significance.

Main modification problems&suggestions:

(1) Ensure the safety of test personnel, add the elderly data in the data set.

(2) It is suggested to add a comparison graph of the readings of the acceleromters when the elderly and young fall.

(3) Explain the meaning of the red line at the bottom of Figure 3.

(4) Some sentences in the text are not clearly expressed. For example, line343.

(5) The core of this article should be the fall detection algorithm, but I can't see any slightly detailed introduction to KNN or SVM algorithms. I suggest adding some algorithm introduciton to the article.

(6) The content of the solution part of this article is somewhat confused, It is suggesnted to draw a flow chart.

(7) The content of the Discussion and Future Work part of this article is not refined enough. It is recommended to condense the content to reduce the length.

(8) Ths structure of this article is not clear enough, and the text is rather cumbersome. It is suggestd to refine the overall content of this article.

Reviewer 2 Report

This paper reported a wrist-based fall detection system to detect falls. I have serious concerns about this manuscript in its current form. The major concerns I had for this manuscript are as follows. Due to the nature of these concerns, I do not recommend its publication

1.       I think the sample used in this project is not appropriate. First, the age range of young participants is mainly 20-23 years old, while the age range of elder participants is 77-95 years old. What is the reason that only choosing these two extreme cases instead of picking up a wide range sample (e.g., 20-95, including 30, 40, 50, 60 )? Second, only healthy elder individuals were involved in this assay, so it missed important populations of unhealthy elder individuals that might be more vulnerable to falls.

2.       The writing of this manuscript is not standard. First, it is hard to tell the number/text in Figures 3-10. It is better to increase the font size. Second, the number format is not professional. For example, the total number of combinations: “1.535.625”. Third, the illustration of points is redundant and confusing. Please follow the standard format: Introduction, Result/Discussion, Conclusion, and Methods/Experimental. Forth, the figures not well explained (e.g., Figure 3-4) in the main text should be moved to supplementary information. Fifth, this manuscript might be the authors’ thesis project, but there is no reason to publish this project in a thesis format.

Other minor concerns:

1.       The abbreviations should be defined upon the first usage in the main text. For example, FDS.

2.       The citation format is not appropriate. Please list the names of methods/datasets of corresponding citations. For example, Lines 105-106.

3.       Since the SisFall dataset is unusable for wrist-based study, why the authors still used it as a basis? (line 142)

Round 2

Reviewer 2 Report

The authors have addressed all my concerns. I recommend this publication after minor revision:

1. change "1.535.625" to "1535625" in line 415.